# ^1^H Nuclear Magnetic Resonance (NMR)-Based Cerebrospinal Fluid and Plasma Metabolomic Analysis in Type 2 Diabetic Patients and Risk Prediction for Diabetic Microangiopathy

**DOI:** 10.3390/jcm8060874

**Published:** 2019-06-19

**Authors:** Huan-Tang Lin, Mei-Ling Cheng, Chi-Jen Lo, Gigin Lin, Shu-Fu Lin, Jiun-Ting Yeh, Hung-Yao Ho, Jr-Rung Lin, Fu-Chao Liu

**Affiliations:** 1Department of Anesthesiology, Chang Gung Memorial Hospital, Taoyuan 333, Taiwan; sanctuary12@cgmh.org.tw; 2College of Medicine, Chang Gung University, Taoyuan 333, Taiwan; 3Graduate Institute of Clinical Medical Sciences, College of Medicine, Chang Gung University, Taoyuan 333, Taiwan; 4Metabolomics Core Laboratory, Healthy Aging Research Center, Chang Gung University, Taoyuan 333, Taiwan; chengm@mail.cgu.edu.tw (M.-L.C.); chijenlo@mail.cgu.edu.tw (C.-J.L.); 5Department of Biomedical Sciences, College of Medicine, Chang Gung University, Taoyuan 333, Taiwan; 6Clinical Metabolomics Core Laboratory, Chang Gung Memorial Hospital, Taoyuan 333, Taiwan; giginlin@cgmh.org.tw; 7Department of Medical Imaging and Intervention, Imaging Core Lab, Chang Gung Memorial Hospital, Taoyuan 333, Taiwan; 8Department of Endocrinology and Metabolism, Chang Gung Memorial Hospital, Taoyuan 333, Taiwan; mmg@cgmh.org.tw; 9Division of Trauma, Department of Plastic and Reconstructive Surgery, Chang Gung Memorial Hospital, Taoyuan 333, Taiwan; a12271@cgmh.org.tw; 10Healthy Aging Research Center, Chang Gung University, Taoyuan 333, Taiwan; hoh01@mail.cgu.edu.tw; 11Department of Medical Biotechnology and Laboratory Science, College of Medicine, Chang Gung University, Taoyuan 333, Taiwan; 12Clinical Informatics and Medical Statistics Research Center and Graduate Institute of Clinical Medicine, Chang Gung University, Taoyuan 333, Taiwan; jr@mail.cgu.edu.tw

**Keywords:** type 2 diabetes, microangiopathy, cerebrospinal fluid, metabolomics, nuclear magnetic resonance

## Abstract

Insulin resistance and metabolic derangement are present in patients with type 2 diabetes mellitus (T2DM). However, the metabolomic signature of T2DM in cerebrospinal fluid (CSF) has not been investigated thus far. In this prospective metabolomic study, fasting CSF and plasma samples from 40 T2DM patients to 36 control subjects undergoing elective surgery with spinal anesthesia were analyzed by ^1^H nuclear magnetic resonance (NMR) spectroscopy. NMR spectra of CSF and plasma metabolites were analyzed and correlated with the presence of T2DM and diabetic microangiopathy (retinopathy, nephropathy, and neuropathy) using an area under the curve (AUC) estimation. CSF metabolomic profiles in T2DM patients vs. controls revealed significantly increased levels of alanine, leucine, valine, tyrosine, lactate, pyruvate, and decreased levels of histidine. In addition, a combination of alanine, histidine, leucine, pyruvate, tyrosine, and valine in CSF showed a superior correlation with the presence of T2DM (AUC:0.951), diabetic retinopathy (AUC:0.858), nephropathy (AUC:0.811), and neuropathy (AUC:0.691). Similar correlations also appeared in plasma profiling. These metabolic alterations in CSF suggest decreasing aerobic metabolism and increasing anaerobic glycolysis in cerebral circulation of patients with T2DM. In conclusion, our results provide clues for the metabolic derangements in diabetic central neuropathy among T2DM patients; however, their clinical significance requires further exploration.

## 1. Introduction

Type 2 diabetes mellitus (T2DM) makes up 90–95% of all diabetes and is characterized by chronic hyperglycemia due to impaired target tissue response to insulin (i.e., insulin resistance) and progressive deterioration of pancreatic β-cell function [1]. Diabetes affected more than 425 million people worldwide in 2017, which is 281% higher than that published in the Diabetes Atlas in 2000, and diabetes accounts for 9.9% of all-cause adult mortality worldwide [2]. This global pandemic of T2DM most likely correlates with a sedentary lifestyle, western diet, overconsumption of high-calorie foods, and other risk factors of metabolic syndrome. The chronic hyperglycemia status in diabetic patients is associated with significant long-term sequelae, ranging from macrovascular (coronary artery disease, peripheral artery disease, atherosclerosis, and stroke) to microvascular complications (retinopathy, nephropathy, and neuropathy) [3]. Given the increasing incidence of T2DM and the long-lasting sequelae of diabetic complications, there is a need for reliable identification of T2DM patients with a higher risk for developing complications.

Metabolomics is a high-throughput analytical technology for the identification and quantitation of small metabolites. It serves as a useful tool for risk stratification and for monitoring disease severity to improve diagnostic and therapeutic efficacy [4]. Nuclear magnetic resonance (NMR) spectroscopy is a highly reproducible and highly quantitative metabolomic platform that enables simultaneous assessment of a wide range of metabolites [5]. Metabolic phenotyping provides a clear insight into the underlying metabolic perturbations in diabetes and its associated complications [6]. Recent metabolomic studies in patients with T2DM showed abnormalities in plasma concentration of carbohydrates, branched-chain amino acids (BCAAs) (isoleucine, leucine, and valine), glycine, aromatic amino acids (AAAs) (especially phenylalanine and tyrosine), glutamine, fatty acids, bile acids, phospholipids, glycerophospholipids, and sphingomyelin [7,8]. A meta-analysis of prospective metabolomic studies reported higher levels of BCAAs and AAAs associated with the development of T2DM and an inverse association of T2DM risk with glycine and glutamine levels [9]. Chronic hyperglycemia leads to downstream diabetic complications through activation of reactive oxygen species (ROS), the generation of advanced glycation end products (AGEs), the flux of the polyol pathway, activation of protein kinase C, excess release of inflammatory cytokines, and exaggerated oxidative stress [10,11]. 

The human brain is an energy-consuming organ with 2% of the total body mass but metabolizes 25% of systemic glucose for energy production [12]. The brain is insulin-sensitive, and insulin resistance in T2DM patients produces multiple behavioral and metabolic effects resulting in diminished cognitive performances of memory and information processing [13]. In rodent models of T2DM, brain insulin resistance impairs hippocampal-based cognitive function and neuroplasticity mainly by a combination of reduced insulin transport across the blood-brain barrier (BBB), mitochondrial dysfunction, ROS overproduction, and neuroinflammation [14]. Several cohort studies reported significant alteration of brain metabolism and increased risk of developing cognitive deficits in T2DM patients, with twice the likelihood to have dementia than the general population [15,16]. Hippocampal insulin resistance may, therefore, exemplify a common feature of metabolic derangements and cognitive dysfunction in T2DM, as well as in aging and Alzheimer’s disease [14,17]. Studies using magnetic resonance spectroscopy showed that cerebral metabolites including N-acetyl-aspartate, choline, creatine, myo-inositol, and glutamate were significantly altered in T2DM, suggesting the involvement of perturbed energy metabolism, neurotransmission, and lipid membrane metabolism in the mechanisms of diabetes-induced brain alterations [13]. 

Cerebrospinal fluid (CSF) exchanges metabolites in the central nervous system (CNS); however, transcellular transport between plasma and CSF was limited by the specialized tight junction of BBB due to the absence of fenestrations and low endocytic activity [18]. The BBB has specialized endothelial cells that maintain CNS homeostasis by regulating electrolyte balance, facilitating nutritional transport, and restricting the entry of neurotoxins, immune cells, and pathogens from systemic circulation [19]. Previous studies using T1-weighted MRI in several human cohorts showed impaired BBB integrity in T2DM patients, which could be due to tight junction disruption and increased paracellular permeability [20,21,22]. Therefore, metabolomic profiling of CSF may reflect brain-specific alterations in T2DM compared to plasma, and the combined analysis of CSF and plasma metabolomics may provide more comprehensive pathognomonic information [23]. Therefore, we conducted this prospective cohort study by NMR spectroscopic analysis of CSF and plasma samples from T2DM patients and control subjects to profile the metabolic alterations in the cerebral and peripheral circulation in T2DM.

## 2. Materials and Methods

This prospective cohort study aimed to compare the metabolomic signature of CSF and plasma of T2DM patients with the control subjects and to identify biomarkers for diabetic microangiopathy. This present study was approved by the Institutional Review Board of Chang Gung Medical Foundation, Taiwan (approval number: 201600122A3), and is registered under Clinical Trials Registry (ClinicalTrials.gov Identifier: NCT03725709). Before participation, the study protocol was explained to all patients and written informed consent was obtained from every participant.

### 2.1. Study Population

A total of 80 participants completed the preliminary evaluation and underwent optional spinal anesthesia for elective surgery from June 1, 2016, to August 31, 2018, at Linkou Chang Gung Memorial Hospital, which is a tertiary teaching hospital in northern Taiwan. Of the 80 study subjects, 40 had a history of T2DM and were taking oral hypoglycemic agents (OHA) whereas the other 40 (control subjects) denied a history of T2DM. The study participants were aged between 20 and 70 years, conformed with The American Society of Anesthesiologists (ASA) physical status classification ≤ 3 in T2DM patients or ASA ≤ 2 in control subjects. All study subjects were admitted one day before elective surgeries, including urologic surgery (transurethral lithotomy, transurethral resection of the prostate), orthopedic surgery for lower extremity (hallux valgus correction, debridement), and lower abdominal surgery (herniorrhaphy) and fasted for at least 8 h before blood and CSF sampling. We excluded four control subjects from subsequent analyses because of their fasting blood glucose > 126 mg/dL, which was classified into diabetes according to American Diabetes Association diagnosis criteria [1]. On the other hand, all the fasting blood glucose levels in T2DM patients were above 126 mg/dL, further confirming the group classification. The final cohort in our study included 40 T2DM patients and 36 controls, and their demographic characteristics including age, sex, height, weight, medication history, in-hospital records, and out-of-hospital visits were recorded. The T2DM patients were further divided into subgroups of different microangiopathies according to recent medical records based on the following definitions: (1) Diabetic retinopathy defined by the presence of non-proliferative diabetic retinopathy or proliferative diabetic retinopathy in retinal photography, and the diagnosis verified by ophthalmologists. (2) Diabetic nephropathy defined by the presence of macroalbuminuria (urinary albumin/creatinine ratio > 300 mg/g) in two spot urine measurements within six months of sampling date, or existence of stage 3–5 chronic kidney disease (estimated glomerular filtration rate (eGFR) < 60 mL/min/1.73 m^2^). (3) Diabetic neuropathy defined by a significant peripheral sensory defect (pinprick, temperature, or vibration sensation) with medication requirement, and the diagnosis confirmed by endocrinologists or neurologists [24,25,26].

### 2.2. CSF and Blood Sampling Procedures

After obtaining informed consent and the use of non-ethanol disinfectant, the spinal anesthesia was achieved with a 26-gauge spinal needle at L3 and L4 interspace in a lateral decubitus position. Once successful, puncture was confirmed by the outflow of clear CSF, 1.2 mL CSF was collected into a polypropylene tube before injection of local anesthetics into the subarachnoid space. In addition, 4 mL blood samples were collected into ethylenediaminetetraacetic acid (EDTA)-coated tubes 10 min before spinal anesthesia. After sampling, spinal anesthesia was administered, and surgical intervention performed. Blood samples were centrifuged for 5 min at 10,000 rpm at 4 °C to obtain plasma samples, and 0.5 mL of plasma and 0.5 mL of CSF samples were sent for biochemical analyses whereas the other samples were stored as aliquots at −80 °C for subsequent metabolomic analyses. None of the patients reported complications or discomfort during spinal anesthesia and sample collection procedures.

### 2.3. Basic Biochemical Analyses

500 μL of plasma and CSF samples were separately analyzed for biochemical and clinical parameters including plasma glucose, plasma insulin, plasma glycated hemoglobin A1c (HbA1c), CSF glucose, and CSF insulin level. Fasting glucose level was determined using the glucose oxidase assay (Cell Biolabs, San Digo, CA, USA), while the insulin and HbA1c concentrations were determined using the enzyme-linked immunosorbent assay (ELISA) kits provided by Mercodia (Uppsala, Sweden) and Cloud-Clone Corp. (Houston, TX, USA), respectively. Subsequently, degree of insulin resistance was compared with the homeostasis model assessment of insulin resistance index (HOMA-IR), which was derived from fasting glucose (mg/dL) × fasting insulin (mU/L) divided by 405. Other biochemical data, including plasma creatinine, eGFR, and urine albumin/creatinine ratio, were obtained from the latest available laboratory results. 

### 2.4. Sample Preparation for NMR Spectroscopy

The CSF samples were prepared by addition of 630 μL of thawed CSF sample to 70 μL of D_2_O solution [1 mM 3-(Trimethylsilyl) propionic-2, 2, 3, 3-d4 acid (TSP), 3 mM NaN_3_] in an Eppendorf tube. Samples were centrifuged for 15 min at 12,000× *g* at 277 K, after which 600 μL of the supernatant was transferred into an NMR tube. On the other hand, 350 μL of thawed plasma sample was mixed with 350 μL of plasma buffer solution (75 mM Na_2_HPO_4_, 0.08% TSP, 2 mM NaN_3_, and 20% D_2_O), and was centrifuged for 15 min at 12,000× *g* at 277 K. Finally, 600 μL of the supernatant was transferred to the 5 mm SampleJet NMR tube for subsequent analysis.

### 2.5. NMR Spectra Acquisition and Processing

The NMR spectrometer contained a Bruker Avance III HD console combined with a 600MHz magnet (Bruker Biospin GmbH, Rheinstetten, Germany). It was equipped with a 5 mm CryoProbe (^1^H/^13^C/^15^N) and SampleJet system with a cooling rack for keeping samples at 279 K. The NMR data were acquired and processed automatically by Topspin software and IconNMR program (version 3.2.2; Bruker Biospin GmbH, Rheinstetten, Germany). 

The Carr-Purcell-Meiboom-Gill (CPMG) spin-echo pulse sequence with water suppression was set up for data acquisition. A relaxation delay of 4 s and T2 relaxation time of 80 ms were applied to attenuate broad signals from proteins. The spectral window was set to 20 ppm, and the 32 transients were acquired with 64 k data points for CSF and plasma. All NMR spectra were phased and baseline-corrected using Topspin software, then referenced to the doublet of ^1^H α-glucose at 5.23 ppm [27]. After processing, the NMR spectra should meet the criterion of quality control that the line width at half height of lactate resonance at 1.32 ppm was <1.15 Hz. 

After removal of the region corresponding to water (5.10–4.20 ppm), the NMR spectral region (between 9.50 and 0.50 ppm) was segmented into bins with the width of 0.01 ppm. The spectral area of each bin was integrated by AMIX software (version 3.9.14; Bruker Biospin GmbH, Rheinstetten, Germany). The chemical shift regions around the residual water (5.10–4.20 ppm) was excluded for score plots of Orthogonal Projections to Latent Structures -Discriminant Analysis (OPLS-DA). The NMR multivariate data was analyzed using Soft Independent Modeling of Class Analogy (SIMCA-P^+^, version 13.0; Umetrics, Umea, Sweden) software. Mean centering and Pareto scaling were used. 

### 2.6. Metabolite Identification and Statistical Analysis

Each metabolite was identified by comparing the resonant frequencies (chemical shifts) and multiplicity patterns of each metabolite using the Human Metabolome Database (HMDB) or the library of Chenomx NMR Suite 7.1 (Chenomx, Edmonton, Canada) [28]. We used a significance level of 0.05, and a correlation coefficient of ±0.396 was employed as the threshold to select variables with the best correlation with the OPLSDA discriminative scores. In addition, predicted values of the response variable Y from the constructed OPLSDA model were used to calculate an area-under-the receiver operating characteristic curve (AUC) value. The metabolites were analyzed and compared for the fold change and AUC value. The other metabolomic analysis such as heatmap and enrichment analysis were accomplished using an online tool MetaboAnalyst 4.0 [29].

Data were presented as means ± SD for continuous variables and as a percentage for qualitative variables (such as sex, medication usage). Statistical analyses were based on NMR signal integration and comparison between two groups was performed using the Student’s *t*-test or χ^2^ tests, and analysis of variance (ANOVA) for comparisons involving multiple groups. The between-group differences of the specific metabolite were identified by OPLSDA coefficients in NMR signals, and the response variable in OPLSDA score plots were compared with goodness of fit (R^2^X, R^2^Y, and Q^2^). In this study, T2DM was our primary outcome, and diabetic microangiopathy (i.e., retinopathy, nephropathy, and neuropathy) was the secondary outcome. The overall scheme for screening metabolites for our primary and secondary outcomes was to identify significant metabolites in CSF and plasma samples that discriminate T2DM and control groups. We constructed multi-marker panels with these metabolites in CSF and plasma samples and selected panels with the highest AUC value as our final panel of biomarkers. It is evident that the diabetic patients with poor glycemic control, hypertension, hyperlipidemia, and obesity are at increased risks of developing diabetic complications, and these factors might influence our measured outcomes of T2DM and diabetic microangiopathy [24,30]. Multivariate logistic regression was applied with adjustment for these confounding factors including age, sex, body mass index (BMI), and medications for chronic diseases (hypertension and hyperlipidemia). All analyses were performed using the SAS software, version 9.4 (SAS Institute Inc. Cary, NC, USA), and two-sided *p* value < 0.05 was considered statistically significant. 

## 3. Results

### 3.1. Demographic Characteristics and Biochemical Parameters

Our final cohort included 40 T2DM patients and 36 control subjects, the flow chart for the study design and group separation is presented in Figure 1. Among the T2DM patients, 14 patients had no documented microangiopathy, 19 patients had diabetic retinopathy, 13 patients had diabetic nephropathy, and the other 11 patients had peripheral neuropathy. The demographic and biochemical parameters for T2DM patients and control subjects are shown in Table 1. According to the medical records and the questionnaire, the control subjects denied other chronic diseases and medication usage with the following exception that two patients had well-controlled hypertension and four patients had well-controlled hyperlipidemia. These control subjects were admitted for elective surgery including transurethral lithotomy (23 patients, 63.8%), hallux valgus correction (11 patients, 30.6%), and herniorrhaphy (2 patients, 5.5%), and all of them were not on additional pre-operative medications. On the other hand, the T2DM patients were significantly higher in age, male percentage, height, weight, and BMI compared to the control group. Besides, more T2DM patients took medications for chronic diseases such as anti-hypertensive agents (57.5% vs. 5.0%) and lipid-modifying agents (55.0% vs. 11.1%) compared to the control subjects. Due to the significant difference between T2DM patients and control subjects, subsequent metabolomic analyses were adjusted for age, sex, BMI, and medications for chronic diseases (hypertension and hyperlipidemia). The median disease duration from the T2DM diagnosis among the T2DM patients was 8.7 ± 5.5 years. All T2DM patients took OHA, and 18 (45%) patients received additional insulin injection for diabetic control. Biochemical analyses showed that the T2DM patients had significantly higher fasting plasma glucose, HbA1c, HOMA-IR, creatinine, and CSF glucose levels compared to the control group. The CSF/plasma ratios of glucose and insulin were 50.45% and 4.92% in T2DM patients and 59.00% and 5.13% in control group.

### 3.2. OPLSDA Score Plots and the OPLS-DA Coefficients of NMR Signals

The OPLSDA score plots for the NMR spectroscopic analysis in CSF and plasma samples from T2DM patients and control subjects are shown in Figure 2. The OPLSDA score plots show clear discrimination between T2DM and control group in CSF samples (reliability: *R*^2^X = 0.757, *R*^2^Y = 0.852, Q^2^ = 0.677) and in plasma samples (reliability: *R*^2^X = 0.905, *R*^2^Y = 0.904, Q^2^ = 0.733). After excluding the EDTA signal in plasma samples, the OPLS-DA coefficient loading plots of NMR signals in CSF and plasma samples of T2DM cases and controls are shown in Figure 3. The metabolites with significant discrimination between T2DM and control group are specified on the NMR spectra according to their chemical shift.

### 3.3. Metabolomic Characteristics of T2DM Patients and Control Subjects

The comparison of NMR signals in CSF and plasma samples from T2DM patients and controls are listed in Table 2 and Table 3, respectively. The NMR signals in CSF samples from T2DM patients had significantly higher levels of mannose, alanine, glycine, leucine, tyrosine, valine, lactate, 2-hydroxybutyrate (2-HB), pyruvate, and phenylalanine than those from controls, but lower levels of histidine. On the other hand, the NMR signals in plasma samples from T2DM patients showed significantly higher levels of N-acetyl glycoprotein (NAG), phenylalanine, polyunsaturated fatty acid (PUFA) (equal to lipid (CH2-CH=CH)), leucine, acetate, and significantly lower citrate, histidine, alanine, and glutamine levels. The metabolite heatmaps are shown for each metabolite in T2DM patients and controls corresponding to their abundance using Pearson’s correlation coefficients analysis (Appendix A). Regarding diabetic patients with diabetic microangiopathy versus diabetic patients without microangiopathy, their corresponding metabolite heatmaps are shown in Appendix A.

### 3.4. Construction of Multi-Marker Panels for Correlation with T2DM and Diabetic Microangiopathy

To derive markers correlating with our primary outcome of T2DM and the secondary outcome of diabetic microangiopathy, we constructed multi-marker panels using candidate metabolites showing a significant difference between T2DM and control groups in CSF and plasma. Using the Student’s *t*-test and stepwise ANOVA, we selected panels with the highest AUC value for correlation with our outcomes. For our primary outcome of T2DM, we selected multi-marker panels constructed from significant metabolites in CSF and plasma with the highest AUC value for correlation with the presence of T2DM and compared to CSF glucose, plasma glucose, plasma HbA1c, and HOMA-IR (Table 4). To exclude possible confounders, we also applied multivariate analysis to compare the correlating odds ratio (OR) of significant metabolites in CSF and plasma for the presence of T2DM by adjusting for age, sex, BMI, and medications for chronic diseases (hypertension and hyperlipidemia) (Appendix A). Inclusion of more metabolites for correlating T2DM resulted in a higher AUC than with a single biomarker, and the selected panels all had a higher AUC value compared to clinical parameters. Notably, the insulin resistance indicator HOMA-IR showed exceptionally high OR compared to any other single parameter. Among the selected panels, a combination of alanine, glycine, histidine, mannose, and pyruvate had the highest AUC of 0.999 and an OR of 4.364 (*p* < 0.05) for identifying T2DM in CSF samples, whereas a combination of acetate, citrate, histidine, leucine, and NAG had the highest AUC of 0.907 and an OR of 5.355 (*p* < 0.05) for identifying T2DM in plasma samples.

For our secondary outcome, we selected five-marker panels of metabolites with significant changes in CSF and plasma that correlate with diabetic retinopathy, nephropathy, and neuropathy among T2DM patients in comparison with the clinical parameters (Table 5). Generally, the selected CSF metabolites had a higher AUC value for correlation with diabetic microangiopathy than plasma metabolites except in neuropathy, and they all exhibited a higher correlation with diabetic microangiopathy than clinical parameters.

For further analysis, we manually identified a six-marker panel of common metabolites that not only significant altered in CSF samples (fold change >1.2 or <0.8, and adjusted *p* value <0.05) but also present in plasma, and examined its correlation with microvascular complications. The six-metabolite combination identified was alanine, histidine, leucine, pyruvate, tyrosine, and valine. Table 6 lists multi-marker panels of metabolites in six-metabolite combination with the highest AUC value or most distinct AUC value for discriminating diabetic retinopathy, nephropathy, and neuropathy. In comparison with other multi-marker panels, the six-metabolite combination showed an excellent correlation for T2DM (AUC of 0.951 in CSF and AUC of 0.879 in plasma). It also showed higher discrimination for diabetic retinopathy (AUC of 0.858 in CSF and AUC of 0.836 in plasma), diabetic nephropathy (AUC of 0.811 in CSF and AUC of 0.865 in plasma), and diabetic neuropathy (AUC of 0.691 in CSF and AUC of 0.822 in plasma). These findings suggest that the six-metabolite combination has the potential to predict future development of diabetic microangiopathy in T2DM patients. Figure 4 illustrates the receiver operating characteristic (ROC) curves of the six-metabolite combination for correlating the presence of T2DM and diabetic microangiopathy. To eliminate possible confounders, we applied multivariate analysis to evaluate the association of each metabolite in six-metabolite combination with the presence of T2DM and diabetic microangiopathy by adjusting for age, sex, BMI, and medications for chronic diseases (hypertension and hyperlipidemia) (Table 7). After adjusting for confounders, the six-metabolite combination still showed significant association with diabetic retinopathy and nephropathy both in CSF and plasma, except for diabetic neuropathy. Besides, tyrosine levels showed a unique association with diabetic microangiopathy in comparison to any other single metabolite. 

### 3.5. Enrichment Analysis and Involved Metabolic Pathways

Altered metabolic pathways for discriminating T2DM were explored by enrichment analysis for metabolites with a significant change in CSF and plasma of T2DM patients, and the results are shown in Appendix A. Among the altered pathways, methyl-histidine metabolism showed significant fold changes both in CSF and plasma samples, and pathway analysis showed that most altered pathways had a connection with the tricarboxylic acid (TCA) cycle. Figure 5 illustrates the metabolic alterations of metabolites in the six-metabolite combination based on glycolysis and the TCA cycle and highlights the proposed connections between metabolites with significant change. After separation of significant metabolites into glucogenic amino acids or ketogenic amino acids, significant increase or decrease of metabolite levels in CSF and plasma samples of T2DM patients were highlighted. Most metabolites showed a similar direction of change in CSF and plasma samples; however, alanine levels showed a significant increase in CSF but a decrease in plasma.

## 4. Discussion

This prospective cohort study profiled the metabolomic signature of T2DM and diabetic microangiopathy in CSF and plasma by ^1^H NMR spectroscopy. NMR metabolomic profiling of fasting CSF samples revealed significantly higher alanine, leucine, tyrosine, valine, lactate, pyruvate, and lower histidine levels. On the other hand, profiling of fasting plasma in T2DM showed higher levels of NAG, phenylalanine, PUFA, leucine, and lower levels of citrate, histidine, alanine, and glutamine. The six-metabolite combination of alanine, histidine, leucine, pyruvate, tyrosine, and valine displayed superior correlation with T2DM in both CSF and plasma samples compared to clinical parameters such as HbA1c, HOMA-IR. It also showed a significant association with diabetic microangiopathy after adjusting for possible confounders. Tyrosine levels showed a unique correlation with diabetic microangiopathy in T2DM patients compared to any other single marker. The CSF metabolome in T2DM patients revealed more significant metabolic alterations than plasma metabolome, implicating higher oxidative stress in brain circulation and possibly underlying BBB dysfunction in T2DM. Further large-scale metabolomic validation of our novel findings and quantification for disease progression is warranted. 

In our results, the identified six-metabolite combination showed high correlation with T2DM and diabetic microangiopathy both in CSF and plasma samples, and is mainly composed of BCAAs (leucine, valine), AAAs (tyrosine), glucogenic amino acids (alanine, histidine), and intermediates of glycolysis (pyruvate). These metabolites are highly associated with insulin resistance as shown in the literature. Insulin resistance is a core feature of T2DM and leads not only to chronic hyperglycemia that defines diabetes, but also to hyperlipidemia, mitochondrial dysfunction, inflammation, oxidative stress, decreased neurotrophic factors, atherosclerosis, and associated metabolic dysfunction [31]. Significant alterations in plasma metabolism in T2DM are anaerobic glycolysis, the TCA cycle, amino acid metabolism, lipogenesis, bile acid metabolism, and fatty acid oxidation [32]. Increase in circulating BCAAs is linked to obesity, insulin resistance, T2DM, and other manifestations of metabolic syndrome [32]. Several factors contribute to an increase in plasma BCAAs during insulin resistance, such as increasing BCAA production from gut microbiota, decreasing BCAA catabolism in adipose tissues, and decreasing liver catabolism by decreased hypothalamic insulin signaling. Subsequently, a further increase in insulin resistance occurs because accumulated BCAAs enhance the incomplete lipid oxidation in muscle, resulting in mitochondrial dysfunction, and substrate flux into lipogenesis [32]. A large-scale human genetic study using Mendelian randomization has verified the causal relationship for the elevation of plasma BCAAs levels and T2DM [33]. Therefore, plasma BCAAs are recognized as a useful biomarker for early detection of insulin resistance and later risk of T2DM, with higher predictive value in men and Asian ethnic groups [33,34,35,36]. Circulating AAAs, especially phenylalanine and tyrosine, were also found to increase in individuals with obesity and insulin resistance [34]. Increases in both BCAA and AAA levels in normoglycemic individuals during long-term follow-up were identified in several large cohorts to have high predictive accuracy for the future development of T2DM, suggesting that dampened metabolism of BCAAs and AAAs play a pivotal role in insulin resistance and the development of T2DM [36,37]. The above evidence supports our results indicating that an increase in BCAAs and AAAs both in CSF and plasma samples demonstrate a superior correlation with T2DM. 

Alanine and histidine are glucogenic amino acids that participate in gluconeogenesis for energy generation, and their plasma levels are lower in T2DM patients during fasting status and cell starvation [38]. Histidine is a precursor of the free-radical scavenger carnosine, and it gained recognition as an anti-oxidant marker for anti-inflammatory and anti-oxidant properties [18,39]. Studies showed that histidine has an inverse association with oxidative stress status in T2DM and its supplementation could improve insulin resistance [40]. The above findings may partly explain the lower plasma alanine and histidine concentrations seen in T2DM patients in our study. Pyruvate is the terminal product of glycolysis, and pyruvate is anaerobically metabolized by lactate dehydrogenase to produce lactate or it can enter the mitochondria for subsequent aerobic oxidative phosphorylation [41]. Therefore, accumulation of pyruvate and lactate in CSF samples suggest an increase in cerebral anaerobic glycolysis and mitochondrial dysfunction that could precipitate cognitive dysfunction in T2DM [42]. Because of these complex relationships with insulin resistance, these metabolites in the six-metabolite combination collectively demonstrated extraordinary correlation with T2DM and its related complications in our study. 

In the present study, we showed a reduction in glutamine levels in CSF and plasma from T2DM patients, which could be due to extensive oxidative stress in T2DM. Glutamate, a downstream product of glutamine, is an important excitatory neurotransmitter and plays a crucial role in incretin/cyclic adenosine monophosphate (cAMP) signaling to increase insulin secretion [43]. In patients with diabetes, chronic glucose hypometabolism and cerebral hypoperfusion could lead to extensive oxidative stress and neuroinflammation [44]. Subsequently, oxidative stress reduces glutamine synthesis and impairs glutamate-glutamine cycle between astrocytes and neurons in T2DM, leading to diabetes-induced neurodegeneration and cognitive dysfunction [45,46]. 

Besides, we also found other significantly altered metabolites in T2DM including citrate, mannose, 2-HB, NAG, PUFA, and acetate. These altered metabolites are involved in hyperglycemia-related cognitive dysfunction, including intermediates of energy metabolism, neurotransmitters, membrane metabolism, and osmoregulation [47]. Citrate is an essential intermediate of the TCA cycle, and decrease in plasma citrate level might imply systemic inhibition of the TCA cycle [42]. The CSF mannose level was significantly higher in T2DM patients in our results. Mannose is involved in glycoprotein biosynthesis and is present at increased levels in patients with T2DM and diabetic retinopathy [25]. Elevated α-hydroxybutyric acid (equal to 2-HB) level is a reliable indicator for impaired glucose tolerance and an early biomarker of diabetes in several cohorts [48]. The increase in plasma levels of PUFA and NAG in T2DM is associated with increased fatty acid oxidation during oxidative stress [38,49]. The plasma acetate level was significantly higher in diabetic patients in previous cohorts, and the production of acetate from pyruvate via acetyl-CoA is increased during fasting status [50]. 

In our analysis, we observed more significant metabolic changes in CSF samples than in plasma samples from T2DM patients, and this cerebral-peripheral discrepancy may implicate higher brain oxidative stress and possible underlying BBB dysfunction in T2DM, but the exact mechanism requires further investigation. Increased alanine and lactate levels in CSF and decreased concentrations in plasma samples from T2DM patients in our study suggest increased glycolysis, decreased gluconeogenesis and oxidative phosphorylation in the brain, while the peripheral tissues exhibit enhanced gluconeogenesis. The above results indicate higher oxidative stress in the brain circulation compared to the peripheral tissues, but its clinical significance requires further validation.

Different diabetic complication may show inconsistent levels of metabolic change, which may serve to diagnose disease progression and triage different complications earlier. In the Action in Diabetes and Vascular Disease-PreterAx and DiamicroN Controlled Evaluation (ADVANCE) trial of T2DM patients with concomitant cardiovascular diseases, lower plasma tyrosine and alanine concentrations were associated with increased risk of microvascular diseases; and higher phenylalanine but lower histidine concentrations were associated with increased macrovascular risks [26]. Somewhat in line with their findings, our analysis showed that a combination of BCAAs, tyrosine, alanine, histidine, and pyruvate could distinguish diabetic patients with microangiopathy from those without microvascular complications. Besides, previous metabolomic studies have identified an association between low tyrosine concentration and diabetic nephropathy, which substantiates our finding that tyrosine level had a significant correlation with diabetic microangiopathy [39]. Because diabetic neuropathy possesses distinct mechanic process from retinopathy or nephropathy, and its mouse model showed most molecular alterations in peripheral nerves instead of in the central ganglia, providing the basis for the lower association of our identified biomarkers with diabetic neuropathy compared to retinopathy or nephropathy [11,51]. 

Many vascular complications present in diabetic patients are due to microvascular compromise [52]. The pathogenesis of diabetes-dependent vascular damage is due to sustained inflammatory and oxidative stress in the endothelial cells promoted by ROS overproduction and AGE activation, which causes a vicious cycle of vascular inflammation, remodeling, and resulting vascular damage [21]. A similar manifestation of cerebral microvascular damage in T2DM might lead to altered BBB transport for many metabolites, as shown in earlier diabetic rat models with increased transport of neutral BCAAs and decreased glucose, insulin, and other basic amino acid transport [21,53]. The higher sensitivity of CSF biomarkers in comparison to plasma metabolites for identification of diabetic microangiopathy in our study may contribute to the prediction of diabetic microangiopathies and provide clues for the pathogenesis of diabetic central neuropathy in T2DM.

To the best of our knowledge, our report is the first to present metabolomic profiling of CSF signature in T2DM patients, and the coincident plasma changes could serve as a cerebral-peripheral comparison for more in-depth insight into insulin resistance in T2DM. CSF is relevant for in vivo sampling for CNS pathology, and metabolic profiling of CSF provides a representative signature of brain metabolism. Earlier studies showed the impact of diabetes on cognitive impairment, mostly in experimental animal models or by using magnetic resonance spectroscopy. Our novel methodology of CSF profiling during routine spinal anesthesia procedure enabled us to probe actual metabolic alterations in the human brain [54]. We also minimized potential bias by excluding four control subjects with potential T2DM. Moreover, the plasma metabolic signature of T2DM in our study was consistent with those in earlier metabolomic studies, thus making our overall results more reliable. The brain insulin resistance and metabolic derangement observed in T2DM patients might serve as potential druggable targets to diminish cognitive dysfunction in T2DM. 

However, there are several limitations to this study. First, the numbers of included T2DM patients and control subjects was small, and significant differences existed in age, sex, BMI, current medications, and chronic diseases. The metabolomic analyses and between-group comparison may be compromised due to the above differences even though we adjusted for these confounders during the comparison. Second, since our included subgroups of diabetic retinopathy, nephropathy, or neuropathy were not mutually exclusive, it was not possible to obtain a unique profile for specific microangiopathy. Third, microvascular complications in our cohort might be under-diagnosed since we defined these subgroups by recent medical records; however, these documented diagnoses make our definition more stringent and thereby making our results more trustworthy. Moreover, in our study, plasma profiling of T2DM was heterogeneous compared to that in earlier publications, which could be due to the heterogeneity in patient selection, biofluid sampling method, as well as technical and population limitations [8,55]. Therefore, future large-scale cohort studies in different populations and disease stages are warranted to validate our novel findings and to examine causal relationships during T2DM progression.

## 5. Conclusions

In this comprehensive metabolomic profiling of CSF and plasma samples from T2DM patients and controls, we provided novel insights of metabolic dysregulation in T2DM and discrepancy of insulin resistance between the brain and peripheral circulation. We identified several dysregulated metabolites in CSF and plasma samples from T2DM patients, which are involved in energy metabolism, amino-acid, and lipid metabolism. The metabolic consequences of T2DM might consist of enhanced glycolysis in the CNS, as well as inhibition of the TCA cycle, enhanced the β-oxidation of fatty acids, and higher gluconeogenesis in the peripheral tissues. Furthermore, a combination of alanine, histidine, leucine, pyruvate, tyrosine, and valine gave superior correlation with T2DM and diabetic microangiopathy, which may contribute to future triage of the development of diabetic complications.

## Figures and Tables

**Figure 1 jcm-08-00874-f001:**
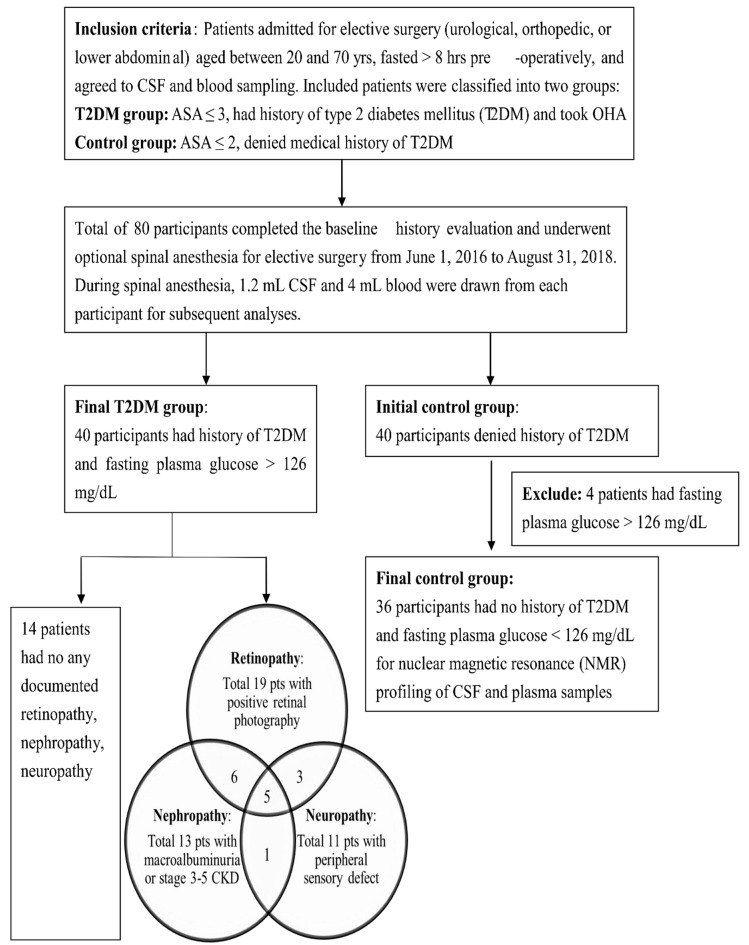
Flow chart for the study design and group separation.

**Figure 2 jcm-08-00874-f002:**
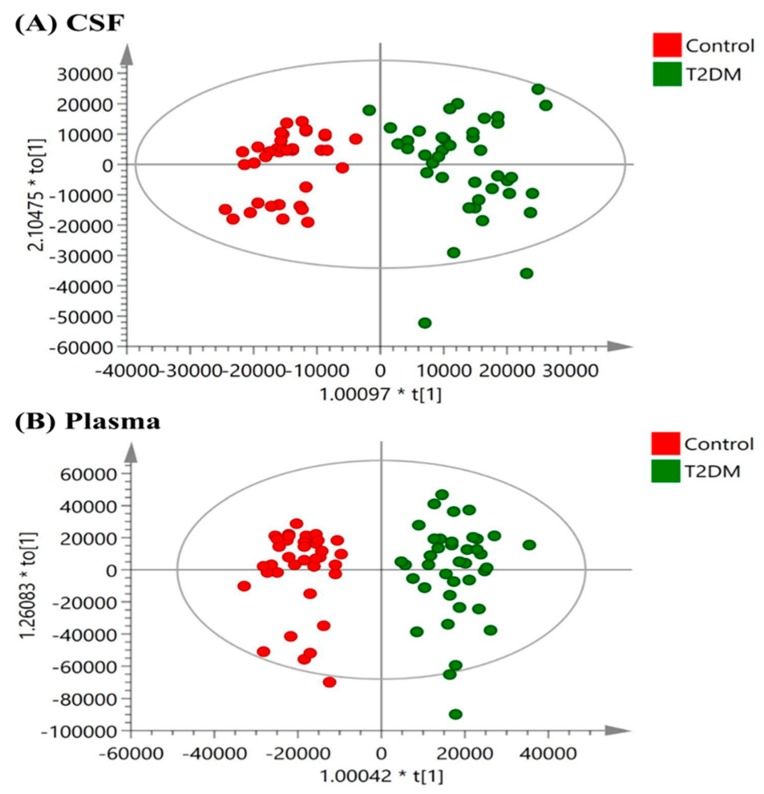
Orthogonal partial least-squares discriminant analysis (OPLSDA) score plots in (**A**) CSF (reliability: *R*^2^X = 0.757, *R*^2^Y = 0.852, Q^2^ = 0.677) and (**B**) plasma (reliability: *R*^2^X = 0.905, *R*^2^Y = 0.904, Q^2^ = 0.733) samples obtained from T2DM patients versus controls. The OPLSDA plots show a clear separation between T2DM and the control group in CSF and plasma samples.

**Figure 3 jcm-08-00874-f003:**
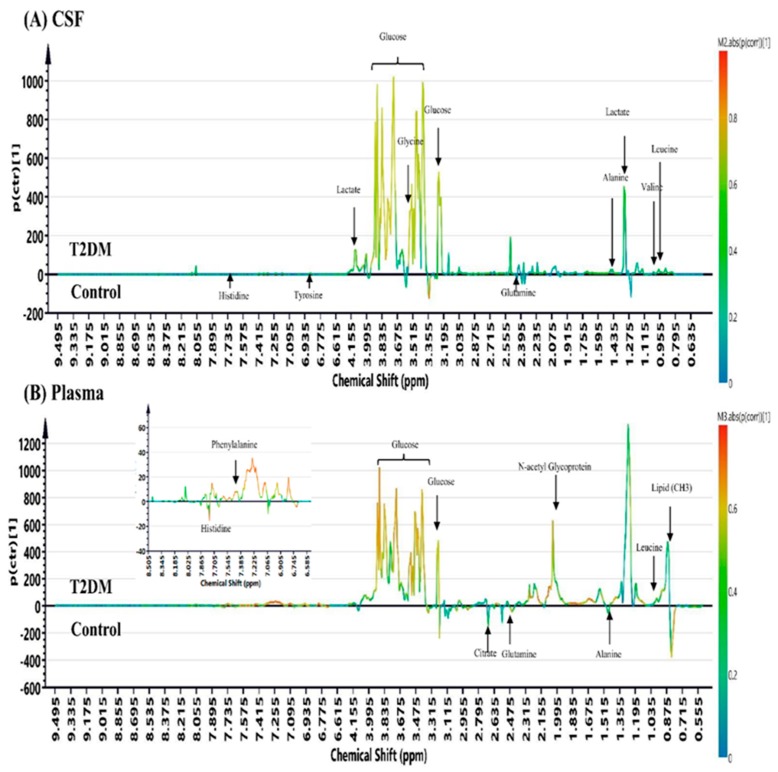
The OPLS-DA coefficient loading plots of NMR signals in (**A**) CSF and (**B**) plasma (excluded EDTA signals) samples obtained from T2DM patients versus control subjects.

**Figure 4 jcm-08-00874-f004:**
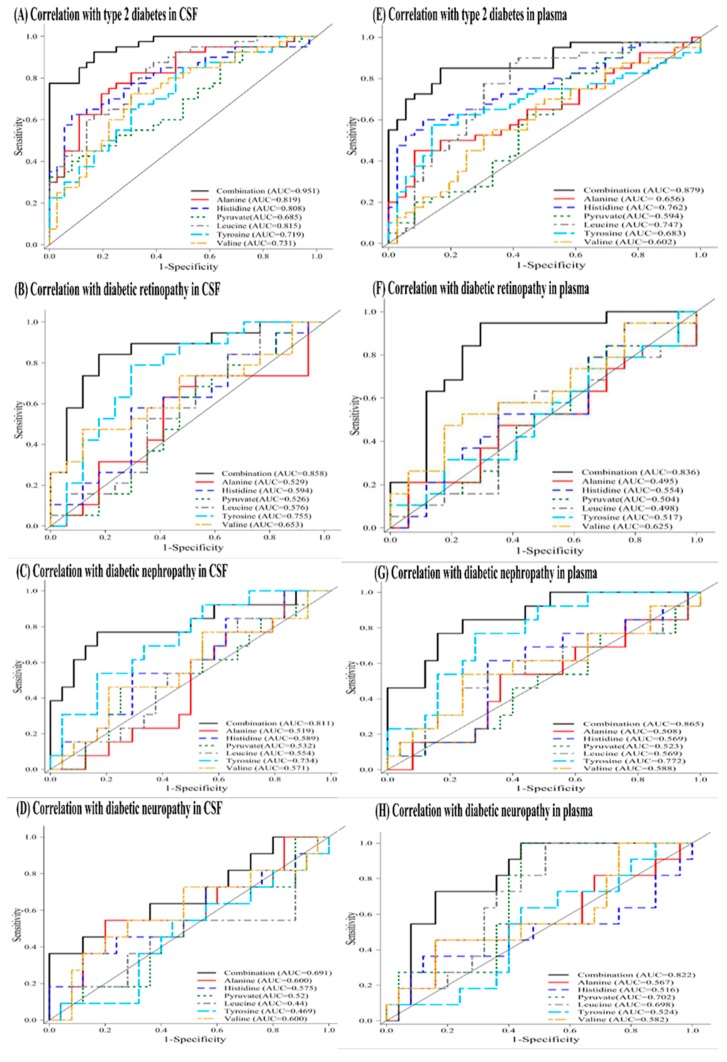
Receiver operating characteristic (ROC) curve of six-metabolite combination for correlation with the presence of type 2 diabetes and diabetic microangiopathy (retinopathy, nephropathy, and neuropathy) in CSF (**A**–**D**) and plasma (**E**–**H**) samples. The six-metabolite combination consists of alanine, histidine, leucine, pyruvate, tyrosine, and valine. Each subfigure consists of seven correlation curves including six single metabolite and their combination.

**Figure 5 jcm-08-00874-f005:**
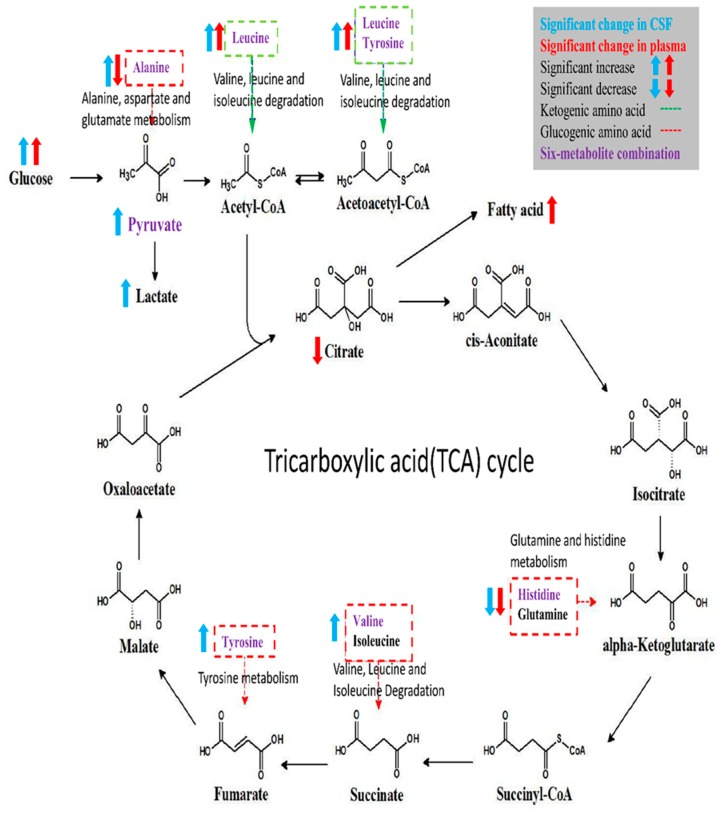
Schematic representation of significant metabolite alterations in CSF and plasma and involved metabolic pathways of six-metabolite combination.

**Table 1 jcm-08-00874-t001:** Demographic characteristics and laboratory parameters in cerebrospinal fluid (CSF) and plasma samples from type 2 diabetes patients and control subjects.

Variables	T2DM (*n* = 40)	Control (*n* = 36)	*p* Value
Male sex, *N* (%)	34 (85.00%)	21 (58.33%)	0.009 *
Age (mean ± SD, years)	55.15 ± 10.46	47.56±14.18	0.009 *
Height (cm)	168.34 ± 6.96	164.71±7.66	0.034 *
Weight (kg)	73.51 ± 12.53	62.75±12.45	<0.001 *
BMI (kg/m^2^)	25.96 ± 4.33	23.03±3.68	0.002 *
**Current medications,** ***N* (%)**			
Oral hypoglycemic agents	40(100%)	N/A	N/A
Insulin injection	18(45%)	N/A	N/A
Anti-hypertensive agents	23(57.5%)	2 (5.5%)	<0.001 *
Lipid modifying agents	22(55%)	4 (11.11%)	<0.001 *
**CSF parameters**			
Fasting glucose (mg/dL)	86.68 ± 31.82	56.38 ± 17.95	<0.001 *
Insulin (mU/L)	0.46 ± 0.53	0.39 ± 0.32	0.563
**Plasma parameters**			
Fasting glucose (mg/dL)	171.82 ± 72.84	95.55 ± 18.26	<0.001 *
HbA1c (%) ^a^	9.63 ± 2.41	5.40 ± 1.61	<0.001 *
Insulin (mU/L)	9.34 ± 5.48	7.59 ± 3.99	0.127
HOMA-IR ^b^	4.24 ± 3.73	1.91 ± 1.23	<0.001 *
Creatinine (mg/dL)	1.20 ± 0.71	0.84 ± 0.30	0.005 *
eGFR (ml/min per1.73m^2^)	87.55 ± 26.64	107.34 ± 27.21	0.002 *
**Complications**			
Diabetic retinopathy ^c^	19(47.5%)	N/A	N/A
Diabetic nephropathy ^d^	13(32.5%)	N/A	N/A
Diabetic neuropathy ^e^	11(27.5%)	N/A	N/A

Abbreviation: T2DM, type 2 diabetes mellitus; BMI, body mass index; HOMA-IR, Homeostatic Model Assessment for Insulin Resistance; HbA1c, glycated hemoglobin A1c; eGFR, estimated glomerular filtration rate; * *p* < 0.05. ^a^ HbA1c was expressed in National Glycohemoglobin Standardization Program (NGSP) unit (%); it would be 82 ± 26 in T2DM and 36 ± 18 in control in International Federation of Clinical Chemistry (IFCC) unit (mmol/mol). ^b^ HOMA-IR = glucose (mg/dL) × insulin (mU/L)/405. ^c^ Diabetic retinopathy definition was the presence of non-proliferative retinopathy (NPDR) or proliferative retinopathy (PDR) in retinal photography as diagnosed by the ophthalmologists. ^d^ Diabetic nephropathy definition was the presence of macroalbuminuria (urinary albumin/creatinine ratio > 300 mg/g) in two spot urine measurements within six months of sampling date, or existence of stage 3–5 chronic kidney disease (eGFR < 60 mL/min/1.73 m^2^). ^e^ Diabetic neuropathy definition was a significant peripheral sensory defect (pinprick, temperature, or vibration sensation) with medication requirement, and the diagnosis confirmed by endocrinologists or neurologists.

**Table 2 jcm-08-00874-t002:** Metabolites changed in CSF samples from T2DM patients versus control subjects.

Metabolites in CSF	Chemical Shift (ppm)	NMR Signal Integration (mean ± SD) (×10^6^ a.u.)		Fold Change	Adjusted *p* Value ^a^
		T2DM	Control	T2DM/Control	T2DM vs. Control
**Increased metabolite levels in T2DM patients compared to controls**					
3-hydroxyisovalerate	1.193	7.72 ± 1.22	3.43 ± 1.30	2.252	0.183
Mannose	5.180	2.73 ± 1.17	1.35 ± 0.48	2.013	<0.001 *
Glycine	.556	4.39 ± 1.35	2.78 ± 0.49	1.579	<0.001 *
α- glucose	5.233	93.61 ± 25.47	62.58 ± 7.38	1.496	<0.001 *
Leucine	0.998	1.32 ± 0.35	0.93 ± 0.27	1.419	<0.001 *
2-hydroxybutyrate	0.893	8.45 ± 4.09	6.33 ± 1.43	1.335	0.043 *
Tyrosine	6.872	1.50 ± 0.49	1.13 ± 0.38	1.330	0.008 *
Alanine	1.455	8.30 ± 2.06	6.26 ± 1.26	1.325	<0.001 *
Valine	1.028	3.32 ± 1.11	2.56 ± 0.67	1.296	0.019 *
Pyruvate	2.366	7.79 ± 2.90	6.03 ± 2.10	1.293	0.044 *
Lactate	4.107	68.48 ± 8.87	58.27 ± 6.59	1.175	0.028 *
Phenylalanine	7.410	0.94 ± 0.32	0.80 ± 0.19	1.170	0.068
Formate	8.450	0.90 ± 0.26	0.79 ± 0.21	1.149	0.095
Isobutyrate	1.065	2.36 ± 0.58	2.08 ± 0.34	1.140	0.289
Creatinine	3.037	17.18 ± 2.59	15.96 ± 2.17	1.076	0.683
Citrate	2.537	22.88 ± 4.00	21.57 ± 4.26	1.061	0.427
Acetate	1.910	6.64 ± 3.11	6.38 ± 3.32	1.040	0.183
**Decreased metabolite levels in T2DM patients compared to controls**					
Glutamine	2.428	47.16 ± 6.44	47.86 ± 4.45	0.985	0.845
Histidine	7.722	0.53 ± 0.14	0.68 ± 0.11	0.779	<0.001 *

^a^*p* value was adjusted for age, sex, body mass index (BMI), and medications for chronic diseases (hypertension, hyperlipidemia); * *p* < 0.05 Abbreviation: T2DM, type 2 diabetes mellitus.

**Table 3 jcm-08-00874-t003:** Metabolites changed in plasma samples from T2DM patients versus control subjects.

Metabolites in Plasma	Chemical Shift (ppm)	NMR Signal Integration (mean ± SD) (×10^6^ a.u.)		Fold Change	Adjusted *p* Value ^a^
		T2DM	Control	T2DM/Control	T2DM vs. Control
**Increased Metabolite levels in T2DM patients compared to controls**					
α-glucose	5.229	109.86 ± 31.31	76.75 ± 11.83	1.431	<0.001 *
Creatinine	4.045	11.11 ± 4.30	8.30 ± 2.62	1.339	0.113
N-acetyl Glycoprotein	2.035	160.20 ± 31.73	120.15 ± 31.28	1.333	<0.001 *
Lipid (CH2-CO)	2.224	33.25 ± 17.18	24.98 ± 14.63	1.331	0.391
Lipid (CH2-CH2-CO)	1.565	99.39 ± 40.21	75.51 ± 36.80	1.316	0.116
Phenylalanine	7.423	3.87 ± 0.89	3.04 ± 0.42	1.273	0.001 *
Lipid (CH2-CH=CH)	1.983	154.88 ± 38.03	129.07 ± 31.12	1.199	0.019 *
Formate	8.452	1.06 ± 0.30	0.88 ±0.27	1.199	0.371
Leucine	0.964	11.18 ± 1.59	9.80 ± 1.74	1.141	0.023 *
Isoleucine	1.009	5.80 ± 1.10	5.14 ± 1.23	1.127	0.277
Tyrosine	6.891	4.13 ± 0.77	3.73 ± 0.52	1.109	0.106
Pyruvate	2.364	7.72 ± 1.94	6.99 ± 1.90	1.105	0.064
Valine	1.035	22.09 ± 3.83	21.11 ± 3.51	1.047	0.885
Lipid (CH3)	0.840	612.38 ± 151.25	596.51 ± 122.31	1.027	0.554
Acetate	1.910	5.28 ± 1.30	5.19 ± 1.26	1.018	0.019 *
**Decreased metabolite levels in T2DM patients compared to controls**					
Lactate	1.322	22.71 ± 6.98	25.07 ± 7.13	0.906	0.870
Histidine	7.765	2.95 ± 0.50	3.30 ± 0.30	0.893	0.001 *
Citrate	2.638	6.04 ± 0.94	6.84 ± 1.06	0.882	<0.001 *
Alanine	1.473	31.66 ± 7.93	35.96 ± 6.46	0.880	0.002 *
Glutamine	2.448	30.78 ± 6.19	35.62 ± 4.31	0.864	<0.001 *

^a^*p* value was adjusted for age, sex, body mass index (BMI), and medications for chronic diseases (hypertension, hyperlipidemia); * *p* < 0.05 Abbreviation: T2DM, type 2 diabetes mellitus.

**Table 4 jcm-08-00874-t004:** Estimated AUC value of multi-marker panels for correlation with the presence of T2DM.

Panels of significantly changed metabolites	AUC for correlation with T2DM
Comparison	T2DM vs. Control
**Panels selected from CSF metabolites**	**AUC for CSF metabolites**
Alanine, Histidine, Mannose	0.985
Alanine, Histidine, Mannose, Pyruvate	0.992
Alanine, Glycine, Histidine, Mannose, Pyruvate	0.999 ^#^
CSF glucose	0.817
**Panels selected from plasma metabolites**	**AUC for plasma metabolites**
Citrate, Histidine, *N*-acetyl-glycoprotein	0.895
Acetate, Citrate, Histidine, *N*-acetyl-glycoprotein	0.907
Acetate, Citrate, Histidine, Leucine, *N*-acetyl-glycoprotein	0.907 ^#^
Plasma glucose	0.896
Plasma HbA1c	0.731
Plasma HOMA-IR	0.750

Abbreviation: T2DM, type 2 diabetes mellitus; AUC, area under the curve estimations; HbA1c, glycated hemoglobin A1c; HOMA-IR, Homeostatic Model Assessment for Insulin Resistance. ^#^ The panel with the highest AUC among the compared category.

**Table 5 jcm-08-00874-t005:** Estimated AUC value of five-marker panels for correlation with diabetic microangiopathy among T2DM patients.

Five-Marker Panels of Significantly Changed Metabolites	Diabetes	Diabetic Retinopathy	Diabetic Nephropathy	Diabetic Neuropathy
Comparison	T2DM vs. Control	T2DM Patients with vs. without Microangiopathy		
**Panels selected from CSF metabolites**	**AUC for CSF**			
Alanine, Leucine, Pyruvate, Tyrosine, Valine	0.889	0.863 ^#^	0.763	0.665
Histidine, 2-hydroxybutyrate, Mannose, Glycine, Valine	0.981	0.734	0.897 ^#^	0.647
Alanine, Histidine, Mannose, Leucine, Tyrosine	0.989	0.814	0.795	0.716 ^#^
CSF glucose	0.817	0.567	0.546	0.636
**Panels selected from plasma metabolites**	**AUC for plasma**			
Alanine, Citrate, Glutamine, Leucine, lipid(CH2-CH=CH)	0.877	0.619 ^#^	0.808	0.753
Acetate, Alanine, Histidine, Leucine, *N*-acetyl-glycoprotein	0.881	0.557	0.812 ^#^	0.738
Acetate, Glutamine, Histidine, Leucine, lipid(CH2-CH=CH)	0.878	0.575	0.760	0.796 ^#^
Plasma glucose	0.896	0.539	0.502	0.669
Plasma HbA1c	0.731	0.505	0.665	0.498
Plasma HOMA-IR	0.750	0.566	0.471	0.687

Abbreviation: T2DM, type 2 diabetes mellitus; AUC, area under the curve estimations; HbA1c, glycated hemoglobin A1c; HOMA-IR, Homeostatic Model Assessment for Insulin Resistance ^#^ The metabolite combinations with the highest AUC among the specified categorical comparison.

**Table 6 jcm-08-00874-t006:** Estimated AUC value for multi-marker panels of metabolites in six-metabolite combination for correlation with diabetic microangiopathy.

Panels of metabolites in Six-Metabolite Combination ^a^	Diabetes	Diabetic Retinopathy	Diabetic Nephropathy	Diabetic Neuropathy
Comparison	T2DM vs. Control	T2DM Patients with vs. without Microangiopathy		
**Panels selected from CSF metabolites with the highest AUC**	**AUC for CSF**			
Alanine, Leucine, Tyrosine, Pyruvate, Valine	0.889	0.836	0.786	0.665
Histidine, Leucine, Pyruvate, Tyrosine, Valine	0.934	0.789	0.808	0.604
Leucine, Pyruvate, Valine	0.837	0.663	0.567	0.672
Six-metabolite combination ^a^	0.951	0.858 ^#^	0.811 ^#^	0.691 ^#^
**Panels selected from plasma metabolites with the highest AUC**	**AUC for plasma**			
Alanine, Histidine, Leucine, Valine	0.863	0.814	0.726	0.753
Histidine, Leucine, Tyrosine, Valine	0.869	0.786	0.837	0.756
Leucine, Pyruvate, Tyrosine, Valine	0.806	0.731	0.825	0.816
Six-metabolite combination ^a^	0.879	0.836 ^#^	0.865 ^#^	0.822 ^#^
**Panels selected from CSF metabolites with the most distinct AUC**	**AUC for CSF**			
Alanine, Valine	0.819	0.783	0.623	0.589
Histidine, Pyruvate, Tyrosine	0.867	0.717	0.805	0.585
Leucine, Pyruvate, Valine	0.837	0.663	0.567	0.672
**Panels selected from plasma metabolites with the most distinct AUC**	**AUC for plasma**			
Alanine, Leucine, Valine	0.821	0.765	0.653	0.746
Alanine, Tyrosine, Valine	0.781	0.625	0.852	0.586
Alanine, Histidine, Leucine, Pyruvate	0.859	0.573	0.597	0.811

Abbreviation: T2DM, type 2 diabetes mellitus; AUC, area under the curve estimations ^#^ The metabolite combinations with the highest AUC among the specified comparison category ^a^ Six-metabolite combination is identified by metabolites with significant change in CSF samples and present in plasma. The identified six-metabolite combination is alanine, histidine, leucine, pyruvate, tyrosine, and valine.

**Table 7 jcm-08-00874-t007:** Association of metabolites in six-metabolite combination with the presence of T2DM and diabetic microangiopathy.

Multivariate Analysis ^b^	T2DM		Diabetic Retinopathy		Diabetic Nephropathy		Diabetic Neuropathy	
**Metabolite in CSF**	**Adjusted OR ^b^**	***p***	**Adjusted OR ^b^**	***p***	**Adjusted OR ^b^**	***p***	**Adjusted OR ^b^**	***p***
CSF glucose	1.061	0.002 *	0.992	0.523	0.979	0.175	0.987	0.511
Alanine	1.009	0.002 *	1.001	0.458	1.000	0.888	1.002	0.345
Histidine	0.871	0.001 *	0.992	0.782	1.064	0.078	1.060	0.183
Leucine	1.039	0.004 *	0.988	0.309	0.989	0.343	1.002	0.893
Pyruvate	1.003	0.049 *	0.999	0.519	0.999	0.584	1.001	0.534
Tyrosine	1.029	0.003 *	0.974	0.017 *	0.975	0.041 *	0.997	0.772
Valine	1.008	0.037 *	0.995	0.225	0.998	0.605	1.002	0.658
Six-metabolite combination ^a^	2.654	0.002 *	2.691	0.004 *	3.012	0.029 *	3.282	0.095
**Metabolites in plasma**	**Adjusted OR ^b^**	***p***	**Adjusted OR ^b^**	***p***	**Adjusted OR ^b^**	***p***	**Adjusted OR ^b^**	***p***
Plasma glucose	1.092	0.003 *	1.003	0.548	0.992	0.172	0.988	0.273
Plasma HbA1c	1.288	0.019 *	0.989	0.855	1.097	0.189	1.028	0.687
Plasma HOMA-IR	1.712	0.027 *	1.063	0.600	0.878	0.292	0.880	0.650
Alanine	0.998	0.004 *	1.000	0.929	1.000	0.829	0.999	0.513
Histidine	0.974	0.003 *	0.998	0.832	1.008	0.329	1.004	0.712
Leucine	1.004	0.036 *	0.998	0.550	0.995	0.080	0.994	0.147
Pyruvate	1.003	0.051	0.914	0.083	0.998	0.385	0.997	0.349
Tyrosine	1.010	0.048 *	1.003	0.628	0.984	0.014 *	1.004	0.677
Valine	1.000	0.911	0.998	0.079	0.999	0.169	0.998	0.214
Six-metabolite combination ^a^	3.352	<0.001 *	2.954	<0.001 *	3.650	<0.001 *	3.237	0.087

Abbreviation: T2DM, type 2 diabetes mellitus; OR, odds ratio; HbA1c, glycated hemoglobin A1c; HOMA-IR, Homeostatic Model Assessment for Insulin Resistance, ^a^ Six-metabolite combination is alanine, histidine, leucine, pyruvate, tyrosine, and valine, ^b^ Adjusted for age, sex, body mass index (BMI), and medications for chronic diseases (hypertension and hyperlipidemia), * *p* < 0.05.

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
