# Peer review of "1H Nuclear Magnetic Resonance (NMR)-Based Cerebrospinal Fluid and Plasma Metabolomic Analysis in Type 2 Diabetic Patients and Risk Prediction for Diabetic Microangiopathy"

_jcm, 2019, doi:10.3390/jcm8060874_

Reviewer 1 Report

Did the participants not suffer from other chronic diseases that could influence the results of the study? I didn't find this data in the manuscript.

Author Response

Point 1: Did the participants not suffer from other chronic diseases that could influence the results of the study? I didn't find this data in the manuscript.

Response 1: We thank the reviewer for kind comment. In fact, we have considered the possible influence of chronic diseases and listed the percentage of current medications for chronic diseases (anti-hypertensive agents, lipid-lowering agents) in T2DM group and control group in Table 1 (please refer to Table 1 in the Result section, page 7). The included control subjects were relatively healthy with only two patients had well-controlled hypertension and four patients had well-controlled hyperlipidemia. Patients in T2DM group had higher percentage of chronic diseases with twenty-three patients took anti-hypertensive agents and twenty-two patients took lipid-lowering agents.

Evidences suggest that diabetic patients with poor glycemic control, hypertension, hyperlipidemia, and obesity are at increased risks of developing diabetic complications [reference], and these factors might influence our measured outcomes of T2DM and diabetic microangiopathy. Since these medications for chronic diseases might have impacts on our measured outcomes, adjustment for chronic diseases might be required. Therefore, we have made adjustment for age, sex, BMI, and chronic diseases (hypertension and hyperlipidemia) for these measured outcomes of T2DM and diabetic microangiopathy and revised Table 2, 3, 4, 5, 6, 7, S1; Figure 4, 5; and related paragraphs in Method section, Result section, and Discussion section.  There are a few changes in the significant metabolites in CSF (with pyruvate and phenylalanine added; and isobutyrate deleted), in significant metabolites in plasma (with acetate added), and in six-metabolite combination (lactate replaced with pyruvate). Despite these changes, the conclusion remains virtually the same. These changes made in the revised manuscript have greatly improved the overall quality of our study.

 Reference:

Marzona, I.; Avanzini, F.; Lucisano, G.; Tettamanti, M.; Baviera, M.; Nicolucci, A.; Roncaglioni, M.C.; Risk; Prevention Collaborative, G. Are all people with diabetes and cardiovascular risk factors or microvascular complications at very high risk? Findings from the Risk and Prevention Study. Acta Diabetol 2017, 54, 123-131, doi:10.1007/s00592-016-0899-0.

Reviewer 2 Report

Authors describe NMR metabolomics analysis conducted by using cerebrospinal fluid (CDF) and plasma to correlate metabolic signatures for T2DM and associated microangiopathy (retinopathy, nephropathy, and neuropathy). A panel of altered CSF and plasma metabolites were identified to correlate with T2DM and associated conditions.

This research work has the merit for publication. However, the study design has major flaws.

Major concerns: 

Authors describe a study that includes patients with T2DM and patients that had no T2DM. However, a few control subjects had conditions other than T2DM indicative with the use of anti-hypertensive and hyperlipidemic drugs (Table 1). In addition, the control subjects had other health conditions, hence they underwent surgery.

1. It is known.that any health condition is reflected by subjects metabolic signature. It may create an experimental bias by including control subjects with other health condition. How could the authors explain to rule out this bias? Did the authors match subjects for medications etc. between the study groups?

2. How do the authors explain the metabolic signatures described in the manuscript are not due to the effects of medications? 

3. Also, it is known that there is variability in the response to drugs among the population. Did the authors consider this fact?.

4. Authors should describe the metabolomics workflow in clear consecutive steps:

For data processing, it should be 1) the removal of unwanted regions (residual water), 2) binning, 3) multivariate data analysis, 4) identifying bins important for discrimination of phenotypes,5) metabolite identification. Authors should describe the multivariate analysis procedure clearly under methods.

5. It is not clear to me what quality control (QC) criteria are followed to assess the quality of metabolomics data.

6. Authors describe Figure 3 as "Representative 1H NMR spectra in (A) CSF and (B) plasma". However, I see that this is a plot of correlation coefficients. ?otherwise, how is it possible to have negative peaks in the spectrum for metabolites?

Minor concerns:

Authors should clearly say ho the metabolites were identified. 

"Resonant frequencies of each metabolite were identified manually by comparing mass spectra and 188 retention time with reference standards from the Human Metabolome Database (HMDB) or the 189 library of Chenomx NMR Suite 7.1 (Chenomx, Edmonton, Canada) [28]." - I think that the authors mixed up the metabolite id procedure, one for mass spectrometry using HMDB, and the other one for NMR (Chenomx).

It is better to organize the heatmap sorted by the phenotype.(it is possible to do it in Mataboanalyst).

Author Response

Comment: Authors describe NMR metabolomics analysis conducted by using cerebrospinal fluid (CSF) and plasma to correlate metabolic signatures for T2DM and associated microangiopathy (retinopathy, nephropathy, and neuropathy). A panel of altered CSF and plasma metabolites were identified to correlate with T2DM and associated conditions. This research work has the merit for publication. However, the study design has major flaws.

Response: We thank the reviewer for kind comment that our research work has the merit for publication. We have re-done our statistical analyses so as to improve the overall quality of the revised manuscript.

Major concerns: Authors describe a study that includes patients with T2DM and patients that had no T2DM. However, a few control subjects had conditions other than T2DM indicative with the use of anti-hypertensive and hyperlipidemic drugs (Table 1). In addition, the control subjects had other health conditions, hence they underwent surgery.

Response: We thank the reviewer for kind comment. According to IRB policy of our institution, it is unethical to recruit the healthy volunteers to receive an unnecessary invasive procedure in human study. We initially planned to include control subjects without systemic diseases in the control cohort. However, the final included control subjects are not completely healthy. According to the medical records and questionnaire, the control subjects denied other chronic diseases and medication usage with the following exception. Two patients had well-controlled hypertension and four patients had well-controlled hyperlipidemia. These control subjects were admitted for elective surgery including transurethral lithotomy (23 patients, 63.8%), hallux valgus correction (11 patients, 30.6%), and herniorrhaphy (2 patients, 5.5%), these etiologies leading to these surgical requirements had little impacts on overall health and had no requirements for long-term medication.

To rule out any effect of chronic diseases and long-term medications on the metabolomic signature, we made additional adjustment for these factors to calculate adjusted p value and odd ratio in the revised manuscript. Therefore, we made adjustment for age, sex, BMI, and chronic diseases (hypertension and hyperlipidemia) for these measured outcomes of T2DM and diabetic microangiopathy, and revised Table 2, 3, 4, 5, 6, 7, S1; Figure 4, 5; and related paragraphs in Method section, Result section, and Discussion section. There are a few changes in the significant metabolites in CSF (with pyruvate and phenylalanine added; and isobutyrate deleted), in significant metabolites in plasma (with acetate added), and in six-metabolite combination (lactate replaced with pyruvate). Despite these changes, the conclusion remains virtually the same. These changes made in the revised manuscript have greatly improved the overall quality of our study.

Point 1. It is known that any health condition is reflected by subject’s metabolic signature. It may create an experimental bias by including control subjects with other health condition. How could the authors explain to rule out this bias? Did the authors match subjects for medications etc. between the study groups?

Response 1: We thank the reviewer for kind comment. Chronic diseases such as hypertension and hyperlipidemia might influence the development of T2DM and diabetic complications; hence, we have adjusted for the medications for chronic diseases (hypertension and hyperlipidemia) between T2DM and control groups and changed related paragraphs in our revised manuscript.

According to the medical records and questionnaire, the control subjects denied other chronic diseases and medication usage with the following exception. Two patients took anti-hypertensive agents (including ARB, CCB) and four patients took lipid-lowering agents (statins). Patients in T2DM group had higher percentage of chronic diseases with twenty-three patients took anti-hypertensive agents (including ARB, CCB, beta-blocker, and diuretics) and twenty-two patients took lipid-lowering agents (including statins and fibrates). As some studies suggest that diabetic patients with poor glycemic control, hypertension, hyperlipidemia, and obesity are at increased risks of developing diabetic complications [reference], we have adjusted for the age, sex, BMI, medications for chronic diseases (hypertension and hyperlipidemia) in our analyses.

Reference:

Marzona, I.; Avanzini, F.; Lucisano, G.; Tettamanti, M.; Baviera, M.; Nicolucci, A.; Roncaglioni, M.C.; Risk; Prevention Collaborative, G. Are all people with diabetes and cardiovascular risk factors or microvascular complications at very high risk? Findings from the Risk and Prevention Study. Acta Diabetol 2017, 54, 123-131, doi:10.1007/s00592-016-0899-0.

Point 2. How do the authors explain the metabolic signatures described in the manuscript are not due to the effects of medications?

Response 2: We thank the reviewer for kind comment. Medications might change related metabolic pathway and influence the measured metabolic signatures. The included patients in our study were all admitted for elective surgery without requirements of additional pre-operative medications (anesthetics and prophylactic antibiotics were given after plasma and CSF sampling). Therefore, medication usage that might influence our analyses were their chronic medications. To rule out the effect of these confounding factors, we have adjusted for the medication factor in our bioinformatic analyses to obtain our profiled metabolic signatures of T2DM and diabetic microangiopathy.

Point 3. Also, it is known that there is variability in the response to drugs among the population. Did the authors consider this fact?

Response 3: We thank the reviewer for kind comment. Indeed, considerable inter-individual variation in drug response exists. Besides, factors such as age, sex, and BMI might influence drug metabolism. To avoid the effect of these factors, we have adjusted for the age, sex, BMI, and medication factor in our bioinformatic analyses to obtain our profiled metabolic signatures of T2DM and diabetic microangiopathy in the revised manuscript.

Point 4. Authors should describe the metabolomics workflow in clear consecutive steps: For data processing, it should be 1) the removal of unwanted regions (residual water), 2) binning, 3) multivariate data analysis, 4) identifying bins important for discrimination of phenotypes,5) metabolite identification. Authors should describe the multivariate analysis procedure clearly under methods.

Response 4: We thank the reviewer for kind comment. We have made a detailed description of metabolomics workflow and the multivariate analysis procedure in the Materials and Methods section entitled “2.5. NMR spectra acquisition and processing” and “2.6. Metabolite identification and Statistical analysis” in the revised manuscript. (please see page 4, line 175 to page 5, line 242)

Point 5. It is not clear to me what quality control (QC) criteria are followed to assess the quality of metabolomics data.

Response 5: We thank the reviewer for kind comment. We have added a related description of quality control criteria and the statement now reads “After processing, the NMR spectrum should satisfy the criterion of quality control that the line width at half height had to be < 1.15 Hz for lactate resonances at 1.32 ppm.” (please refer to the Method section, 2.5.NMR spectra acquisition and processing, page 4, line 185-187)

Point 6. Authors describe Figure 3 as "Representative 1H NMR spectra in (A) CSF and (B) plasma". However, I see that this is a plot of correlation coefficients? Otherwise, how is it possible to have negative peaks in the spectrum for metabolites?

Response 6: We thank the reviewer for kind comment. We have corrected the figure legend of the Figure 3 in the revised manuscript. It now reads” OPLS-DA coefficient loading plots of NMR signals in (A) CSF and (B) plasma (excluded EDTA signals) samples obtained from T2DM patients versus control subjects.” (please refer to the Result section, 3.2. OPLS-DA score plots and the OPLS-DA coefficients of NMR signals, page 9, line 299-300)

Minor concerns: Authors should clearly say how the metabolites were identified.

Point 1:"Resonant frequencies of each metabolite were identified manually by comparing mass spectra and retention time with reference standards from the Human Metabolome Database (HMDB) or the library of Chenomx NMR Suite 7.1 (Chenomx, Edmonton, Canada)." - I think that the authors mixed up the metabolite id procedure, one for mass spectrometry using HMDB, and the other one for NMR (Chenomx).

Response 1: We thank the reviewer for kind comment. This NMR metabolomic study used resonant frequencies for metabolite identification. We have corrected the metabolite identification procedure in the revised manuscript. The statement now reads” Resonant frequencies of each metabolite were identified from the Human Metabolome Database (HMDB) or the library of Chenomx NMR Suite 7.1 (Chenomx, Edmonton, Canada).” (please refer to the Method section, 2.6. Metabolite identification and Statistical analysis, page 4, line 198-199)

Point 2: It is better to organize the heatmap sorted by the phenotype. (it is possible to do it in MetaboAnalyst).

Response 2: We thank the reviewer for kind comment. We have corrected Figure S1 (metabolite heatmaps comparing T2DM patients and control subjects), and added Figure S2 (metabolite heatmaps comparing T2DM patients with diabetic microangiopathy versus T2DM patients without microangiopathy) in the revised manuscript. These heat maps are sorted by the phenotype. (please refer to the page 21-22)

Round  2

Reviewer 2 Report

Authors addressed my concerns and revised the manuscript.

Minor comments: Please include the following changes to the manuscript.

Line 176, include 600 MHz instead of 14.1T

Line 181, include cpmg pulse sequence with water suppression (I believe that you did it) instead of cpmg pulse sequence.

Line 188, include "after removing water region (4.79-4.90 ppm)". Please use the correct ppm regions that you used.

You need to write the normalization method (that you used in AMIX software, for exaple total sum normalization to all integrals) in Section 2.5. 

instead of "scaling methos was Pareto scaling", changed it to "mean centering and Pareto scaling was used".

Note: SIMCA software automatically does the mean centering. 

Remove the the text in the two sentences beginning in Line 196 and ending in line 199 add the following text as follows: "Each metabolite was identified by comparing the resonant frequencies (chemical shifts) and multiplicity patterns of each metabolite using the Human Metabolome Database (HMDB) or the library of Chenomx NMR Suite 7.1 (Chenomx, Edmonton, Canada) [28]. 

Note: you do not do the intgral calculation for metabolite identification. It is the chemical shift and multiplicity pattern of the signals. I do not think that you need both HMDB and Chenomx unless you used both.

Line 225, you wrote: "Data were presented as means ± SD for continuous variables and as a percentage for qualitative variables". 

Could you please clarify what these qualitative variables are". Also please indicate that you used NMR signal intensities were used for statistics. 

Note: For relative quantification of metabolites, you need an internal standard with known concentration. You do have an internal standard (TSP). However, its intensity cannot be used because it binds to proteins, hence, broadening the signal..Also, you did not use it for chemical shift referencing either.

In table 2 and table 3, change "NMR Signals" to "NMR Signal Intensity".

Another important reference to include on Branched-chain amino acids and risk of diabetes is Wang et al 2011. Metabolite Profiles and the Risk of Developing Diabetes. Nature Medicine 17(4): 448–453. doi:10.1038/nm.2307.

Please include this reference accordingly.

Author Response

Minor comments:

Point 1. Please include the following changes to the manuscript.

(1)  Line 176, include 600 MHz instead of 14.1T

(2)  Line 181, include cpmg pulse sequence with water suppression (I believe that you did it) instead of cpmg pulse sequence.

(3)  Line 188, include "after removing water region (4.79-4.90 ppm)". Please use the correct ppm regions that you used.

Response 1: We thank the reviewer for kind comment. The corrected water region was 5.10-4.20 ppm. We have revised our manuscript according to your suggestion.

Point 2. You need to write the normalization method (that you used in AMIX software, for example: total sum normalization to all integrals) in Section 2.5.

Response 2: We thank the reviewer for kind comment. We did not perform normalization of NMR spectra in AMIX software. We have revised this sentence as “The spectral area of each bin was integrated by AMIX software” (please refer to the Method section,2.5. NMR spectra acquisition and processing, page 4, line 189-190).

Point 3. instead of "scaling methods was Pareto scaling", changed it to "mean centering and Pareto scaling was used".

(Note: SIMCA software automatically does the mean centering.)

Response 3: We thank the reviewer for kind comment. We have revised this sentence as” Mean centering and Pareto scaling were used.” (please refer to the Method section,2.5. NMR spectra acquisition and processing, page 4, line 194)

Point 4. Remove the text in the two sentences beginning in Line 196 and ending in line 199. Add the following text as follows: "Each metabolite was identified by comparing the resonant frequencies (chemical shifts) and multiplicity patterns of each metabolite using the Human Metabolome Database (HMDB) or the library of Chenomx NMR Suite 7.1 (Chenomx, Edmonton, Canada) [28].

(Note: you do not do the integral calculation for metabolite identification. It is the chemical shift and multiplicity pattern of the signals. I do not think that you need both HMDB and Chenomx unless you used both.)

Response 4: We thank the reviewer for kind comment. We have revised our manuscript according to your suggestion. (please refer to the Method section, 2.6. Metabolite identification and Statistical analysis, page 4, line 196-198)

Point 5. Line 225, you wrote: "Data were presented as means ± SD for continuous variables and as a percentage for qualitative variables". Could you please clarify what these qualitative variables are? Also please indicate that you used NMR signal intensities were used for statistics.

(Note: For relative quantification of metabolites, you need an internal standard with known concentration. You do have an internal standard (TSP). However, its intensity cannot be used because it binds to proteins, hence, broadening the signal. Also, you did not use it for chemical shift referencing either.)

Response 5: We thank the reviewer for kind comment. Qualitative variables are these variables presented as N (%) in Table 1 including sex, medication usage, and numbers of patients with microangiopathy. We used NMR signal integration for statistical analyses. We have revised our manuscript and it now reads “Statistical analyses were based on NMR signal integration and comparison between two groups was performed using the Student’s t-test or χ2 tests, and analysis of variance (ANOVA) for comparisons involving multiple groups.”(please refer to the Method section, 2.6.Metabolite identification and Statistical analysis, page 5, line 238-240)   

Point 6. In table 2 and table 3, change "NMR Signals" to "NMR Signal Intensity".

Response 6: We thank the reviewer for kind comment. We have revised “NMR signals” to “NMR signal integration” in Table 2 and Table 3.

Point 7. Another important reference to include on Branched-chain amino acids and risk of diabetes is Wang et al 2011. Metabolite Profiles and the Risk of Developing Diabetes. Nature Medicine 17(4): 448–453. doi:10.1038/nm.2307. Please include this reference accordingly.

Response 7: We thank the reviewer for kind comment. We have already cited this important article as reference 36 in our original manuscript for providing evidences of combining BCAA and AAA levels in normoglycemic individuals have high predictive accuracy for the future development of T2DM. We have added this important reference in sentences discussing “BCAAs and risk of DM” in our revised our manuscript according to your suggestion.
